# A Novel Multi-Axial Pressure Sensor Probe for Measuring Triaxial Stress States Inside Soft Materials

**DOI:** 10.3390/s21103487

**Published:** 2021-05-17

**Authors:** Giuseppe Zullo, Anna Leidy Silvestroni, Gianluca Candiotto, Andrey Koptyug, Nicola Petrone

**Affiliations:** 1Department of Industrial Engineering, University of Padua, Via Venezia 1, 35131 Padua, Italy; giuseppe.zullo@phd.unipd.it (G.Z.); annaleidy.silvestroni@studenti.unipd.it (A.L.S.); candiotto.gianluca@gmail.com (G.C.); 2Department of Quality and Mechanical Engineering, Mid Sweden University, Campus Östersund Kunskapens väg 8, SE-831 25 Östersund, Sweden; andrey.koptyug@miun.se

**Keywords:** pressure sensor, stress state, shear stress, soft materials, tissue surrogates

## Abstract

This paper presents the concept, design, construction, and validation of a novel probe based on the hexadic disposition of six pressure sensors suitable for measuring triaxial stress states inside bulky soft materials. The measurement of triaxial stress states inside bulk materials such as brain tissue surrogates is a challenging task needed to investigate internal organs’ stress states and validate FE models. The purpose of the work was the development and validation of a 17 × 17 × 17 mm probe containing six pressure sensors. To do so, six piezoresistive pressure sensors of 6 mm diameter were arranged into an hexad at three cartesian axes and bisecting angles, based on the analytical solution of the stress tensor. The resulting probe was embedded in a soft silicone rubber of known characteristics, calibrated under cyclic compression and shear in three orientations, and statically validated with combined loads. A calibration matrix was computed, and validation tests allowed us to estimate Von Mises stress under combined stress with an error below 6%. Hence, the proposed probe design and method can give indications about the complex stress state developing internally to soft materials under triaxial high-strain fields, opening applications in the analysis of biological models or physical surrogates involving parenchyma organs.

## 1. Introduction

In recent years, there has been a growing interest in the application of sensors in soft materials for the development of instrumented biofidelic surrogates. Indeed, such materials are widely used where high flexibility and strain are required, and their instrumentation represents a challenge that could improve the measurement of the human body and wearable devices. Soft materials such as polydimethylsiloxane (PDMS) or other silicone rubbers are investigated as possible soft tissue surrogates [1,2], and brain surrogates made of these materials are adopted by researchers who are trying to improve the biofidelity of their physical models of the human head [3,4,5,6]. A biofidelic head model is needed to improve the evaluation of brain kinematics during impacts and, thus, the effectiveness of head protective gear.

However, despite the number of physical surrogates already developed [3,4,5,6], finite element (FE) models of the head are still superior in terms of measure capability, as the stress state of tissues is given as a continuous field. On this matter, an estimate of brain strain and stress during dynamic impacts is available only with numerical methods [7,8,9] and lacks an experimental counterpart that could validate FE models and improve the comprehension of this phenomena. A measure of brain stress and strain could improve the experimental methods to test the effectiveness of head protective gear, since these quantities are correlated with traumatic brain injury (TBI) such as diffused axonal injury (DAI) a major cause of death after traumatic events [9,10].

Therefore, we focused on the development of a multi-axial pressure sensor (MAPS) probe to enhance the measurement capability of a physical head model and to expand possible solutions to measure stress states inside soft materials. Moreover, the knowledge of the stress state could be paired with material models to obtain the strain state present inside soft materials.

The development of this kind of sensor is challenging due to multiple factors: the high softness of the material reduces the typology of suitable transducers, and the high number of measurement channels that must be condensed in a very limited space limits their size.

To start searching for possible solutions, we conducted a deep literature review, which considered past works related to the stress measurement of soft materials. Many authors developed their flexible strain sensor using PDMS as support [11]. Some researchers embedded silver particles inside a PDMS compound [11,12]. The result of their research was a highly stretchable strain sensor, which was applied to study the motion of anatomical joints. However, the preparation of this sensor appears complex as well as its connection to signal cables. Similarly, other researches involved the use of carbon nanotubes in combination with different materials in films or cylinders [13,14,15]. Wang et al. created and tested one of the first sensors using silicone rubber filled with carbon nanotubes [14]. The group created a square element (10 × 10 × 2 mm) and subjected it to a pressure ranging from 0 to 2 MPa. They showed that a content of filler under 18% in volume is suitable for the creation of a strain sensing element, having a monotonic resistance variation with the possibility of calibrating a sensor. The main advantage of the sensors presented above is that they are almost entirely made of the same material of the surrounding gel, minimizing the distortion of the strain field. However, the ability of resolving multi-axial stress states was not explored by the authors, and still remains a crucial quest. Conversely, the research involving the measure of multiple axes was instead focused on the inclusion of rigid transducers inside a soft material. Laszczak et al. developed and calibrated an interfacial planar stress sensor for prosthetic sockets by disposing conductive foils separated by a dielectric gel, creating a set of planar capacitors capable of sensing the normal and the shear stresses on a plane [16,17]. The realization and wiring of this sensor seems more feasible than the previous, but it is still challenging because the sensor structure must be adapted to the 3D stress, and the conversion of capacitance into voltage values requires external specific hardware, making the data acquisition more complex. Using a similar principle, a shear sensor suitable for wheelchair cushions and prosthetics sockets was also developed in the past by Toyama et al. by using polymeric films filled with a liquid electrolyte [18]. Despite the attractive efficacy of this sensor, its arrangement inside bulky soft material remained unexplored and appears problematic. Using a different approach, Dwivedi et al. developed a magnetic contact 3D force sensor by using a hall effect sensor paired with a magnet embedded in a pyramid of soft silicone rubber [19]. This sensor performed well in detecting contact forces, but the possibility of embedding it inside a bulk material was not explored as well as its expansion in measuring the six components of stress. A triaxial strain state sensor was developed by Francois et al. who created a steel ellipsoidal inclusion instrumented with six sensing fibers, which was embedded inside concrete [20]. Despite the good results obtained, the extremely different mechanical properties of soft materials and the high spatial resolution desired in the study of head impacts make the sensor not suitable for the specific application.

Given the limitation of previous research, we developed a MAPS probe using commercial pressure sensors. This work explored the possibility of evaluating the multi-axial stress of a brain surrogate specimen made of silicone rubber by embedding a hexad of pressure sensors. We decided to use pressure transducers after an evaluation, which considered their affordable cost, the low effort put in their preparation, and their simplicity in signal processing and interpretation. Thanks to a preliminary analytical study, we were able to investigate possible dispositions for the pressure sensors to optimize the size and the sensitivity of the MAPS stress probe, and we managed to realize a specimen that embedded our hexad.

Upon making the stress probe, we conducted calibration trials aiming to relate the pressure sensors outputs with the external loads, creating a calibration matrix. Finally, the stress probe was subjected to bi-axial combined load tests in different orientations to verify its capability of distinguishing and estimating multiple loads correctly. The results of this study confirmed that our MAPS probe is suitable to measure the triaxial stress state, and thus, it can be positioned inside a brain surrogate in a predefined set of meaningful locations to obtain an experimental measure of the triaxial stress state during impact tests.

Further investigations will be conducted to check the repeatability of the sensor and its performances under dynamic combined stress states, to reduce its size, and finally, to evaluate its performances in full-scale helmet testing.

## 2. Materials and Methods

### 2.1. Stress State Analysis and Pressure Sensor Disposition

Stress state in materials can be represented by a three-by-three tensor **σ**, which has six independent components: σ_x_, σ_y_, σ_z_, τ_xy_, τ_yz_, and τ_xz_. Thus, to estimate the triaxial stress state inside a brain surrogate, at least six measure channels are needed to estimate six stress components.

To overcome the fact that pressure sensors can only sense normal stresses acting on their surface while they could not sense shear stresses, we took advantage of the trans-formation law of the stress tensor. Indeed, the components of **σ** vary with respect to the reference system considered for the computation of the tensor. The relation between stress **σ** in the XYZ reference system and the stress **σ’** in the X′Y′Z′ reference system is given by the equation:(1)$$\sigma^{\prime }=A\sigma A^{T}$$
where ***A*** is the orientation matrix constituted by the cosine directors, which change the basis from the XYZ to the X′Y′Z′ reference system.

From the matrix product, we get the expressions linking the three normal stresses in the X′Y′Z′ reference system to the normal and shear stresses in the original reference system:(2)$$\sigma_{x}^{\prime }=a_{11}^{2}\sigma_{x}+a_{12}^{2}\sigma_{y}+a_{13}^{2}\sigma_{z}+2a_{11}a_{12}\tau_{\mathrm{xy}}+2a_{12}a_{13}\tau_{\mathrm{yz}}+2a_{11}a_{13}\tau_{\mathrm{xz}}$$
(3)$$\sigma_{y}^{\prime }=a_{21}^{2}\sigma_{x}+a_{22}^{2}\sigma_{y}+a_{23}^{2}\sigma_{z}+2a_{21}a_{22}\tau_{xy}+2a_{22}a_{23}\tau_{yz}+2a_{21}a_{23}\tau_{xz}$$
(4)$$\sigma_{z}^{\prime }=a_{31}^{2}\sigma_{x}+a_{32}^{2}\sigma_{y}+a_{33}^{2}\sigma_{z}+2a_{31}a_{32}\tau_{xy}+2a_{32}a_{33}\tau_{yz}+2a_{31}a_{33}\tau_{xz}$$
where a_ij_ are the terms of the change in basis matrix ***A***. If we analyze these three equations, we can see that the three normal stresses in the X′Y′Z′ reference system are a linear combination of the six stresses in the original reference system.

To create the stress probe, we placed a first triad of pressure sensors: *p*_1_, *p*_2_, and *p*_3_ aligned to the XYZ axes to get a measure of σ_x_, σ_y_, σ_z_, respectively. Then, we placed three auxiliary sensors: *p*_4_, *p*_5_, *p*_6_ aligned to the bisectors of the XY, YZ, and XZ axes, respectively. This disposition recalls the configuration of the 0, 45, 90° strain gauge rosettes, extended to the three dimensions. In this configuration, when the external load is either σ_x_, σ_y_, σ_z_, τ_xy_, τ_yz_, or τ_xz_, the first principal stress is aligned to the sensing direction of *p*_1_, *p*_2_, *p*_3_, *p*_4_, *p*_5_, or *p*_6_, respectively. The resulting spatial disposition of the pressure sensors is shown in Figure 1.

In this spatial disposition, the three auxiliary sensors do not form a cartesian reference system, but each must be considered as one axis of three separate reference systems. To obtain these reference systems, the original reference system XYZ was rotated by a 45° rotation, respectively. This is formalized by introducing the basic rotation matrices ***A*_X_**, ***A*_Y_**, and ***A*_Z_,** which bring the coordinates of the XYZ reference system to the auxiliary systems. In this case α, β, and γ are equal to 45°, and the three rotations about its X, Y, and Z axes align the auxiliary sensors to the Y′′, Z′′′, and X′ axes, respectively.
(5)$$A_{X}=\left[\begin{array}{ccc} 1 & 0 & 0\\ 0 & cos\left(\alpha \right) & -sin\left(\alpha \right)\\ 0 & sin\left(\alpha \right) & cos\left(\alpha \right) \end{array}\right]=\left[\begin{array}{ccc} 1 & 0 & 0\\ 0 & \sqrt{2}/2 & -\sqrt{2}/2\\ 0 & \sqrt{2}/2 & \sqrt{2}/2 \end{array}\right]$$
(6)$$A_{Y}=\left[\begin{array}{ccc} cos\left(\beta \right) & 0 & sin\left(\beta \right)\\ 0 & 1 & 0\\ -sin\left(\beta \right) & 0 & cos\left(\beta \right) \end{array}\right]=\left[\begin{array}{ccc} \sqrt{2}/2 & 0 & \sqrt{2}/2\\ 0 & 1 & 0\\ -\sqrt{2}/2 & 0 & \sqrt{2}/2 \end{array}\right]$$
(7)$$A_{Z}=\left[\begin{array}{ccc} cos\left(\gamma \right) & -sin\left(\gamma \right) & 0\\ sin\left(\gamma \right) & cos\left(\gamma \right) & 0\\ 0 & 0 & 1 \end{array}\right]=\left[\begin{array}{ccc} \sqrt{2}/2 & -\sqrt{2}/2 & 0\\ \sqrt{2}/2 & \sqrt{2}/2 & 0\\ 0 & 0 & 1 \end{array}\right]$$

The change in basis modifies the equations of stress, and the normal stresses acting on the auxiliary sensors surface can be expressed in function of terms of the three rotation matrices ***A_X_***, ***A_Y_***, and ***A_Z_*** and terms of stresses in the original reference system XYZ:(8)$$\sigma_{x}^{\prime }=a_{Z,11}^{2}\sigma_{x}+a_{Z,12}^{2}\sigma_{y}+a_{Z,13}^{2}\sigma_{z}+2a_{Z,11}a_{Z,12}\tau_{xy}+2a_{Z,12}a_{Z,13}\tau_{yz}+2a_{Z,11}a_{Z,13}\tau_{xz}$$
(9)$$\sigma_{y}^{\prime \prime }=a_{X,21}^{2}\sigma_{x}+a_{X,22}^{2}\sigma_{y}+a_{X,23}^{2}\sigma_{z}+2a_{X,21}a_{X,22}\tau_{xy}+2a_{X,22}a_{X,23}\tau_{yz}+2a_{X,21}a_{X,23}\tau_{xz}$$
(10)$$\sigma_{z}^{\prime \prime \prime }=a_{Y,31}^{2}\sigma_{x}+a_{Y,32}^{2}\sigma_{y}+a_{Y,33}^{2}\sigma_{z}+2a_{Y,31}a_{Y,32}\tau_{xy}+2a_{Y,32}a_{Y,33}\tau_{yz}+2a_{Y,31}a_{Y,33}\tau_{xz}$$

We can substitute in the expressions the known terms given by the measured output of pressure sensors and by their spatial disposition, obtaining the three simplified equations:(11)$$-p_{4}=-p_{1}/2-p_{2}/2-\tau_{xy}$$
(12)$$-p_{5}=-p_{2}/2-p_{3}/2-\tau_{yz}$$
(13)$$-p_{6}=-p_{1}/2-p_{3}/2-\tau_{xz}$$

The three equations above form a system with three variables: τ_xy_, τ_yz_, and τ_xz_. The system was solved obtaining the equations expressing τ in function of the output of pressure sensors. We can finally express the six components of the stress tensor in the function of the output of the pressure sensors.
(14)$$\sigma_{x}=-p_{1}$$
(15)$$\sigma_{y}=-p_{2}$$
(16)$$\sigma_{z}=-p_{3}$$
(17)$$\tau_{xy}=-p_{1}/2+p_{2}/2+p_{4}$$
(18)$$\tau_{yz}=-p_{1}/2+p_{3}/2+p_{5}$$
(19)$$\tau_{xz}=+p_{2}/2+p_{3}/2+p_{6}$$

Thanks to these equations, the stress state of the material in the point of application of the MAPS probe can be resolved, and meaningful parameters such as the Von Mises stress (VMS) could be computed using the equation:(20)$$\sigma_{id,VM}=\sqrt{\sigma_{x}^{2}+\sigma_{y}^{2}+\sigma_{z}^{2}-\left(\sigma_{x}\sigma_{y}+\sigma_{y}\sigma_{z}+\sigma_{x}\sigma_{z}\right)+3\left(\tau_{xy}^{2}+\tau_{yz}^{2}+\tau_{xz}^{2}\right)}$$

With this set of equations, the stress in the point of application of the probe can be fully determined in its components and eventually condensed in the VMS parameter to compare the stress intensity with a threshold.

### 2.2. Soft Material Adopted

To test the MAPS probe, we embedded the sensors inside a cubic specimen made of PlatsilGel OO30 (Polytek Development Corp.). This material consists of a bicomponent (A and B) silicone rubber, which must be mixed in a 1:1 ratio (by weight or volume). To make the material softer, two parts of Deadener (Polytek Development Corp.) were added to the mixture. According to the producer’s technical specification, this procedure reduces the material Shore hardness from the original OO30 to OOO16. Upon mixing the components together, four hours of curing time at room temperature were needed before the gel could be demolded and used. Moreover, this material allows for multiple step castings since uncured material bonds to already cured surfaces.

This material was characterized by means of several mechanical tests, and a material model was fitted to the data to obtain the stress–strain relationship. More details of the characterization procedure and results can be found in Appendix A.

### 2.3. Preparation of Pressure Sensors

The pressure sensor that was adopted for the MAPS probe is the MS5407-AM (TE Connectivity) piezoresistive pressure transducer. This sensor offers a full-scale pressure of 700 kPa and transmits data through a differential analog output (full scale output: 392 mV, 5 V power supply); according to the producer’s specifications, linearity is ±0.15% (typical) of the full-scale pressure. In terms of overall sizes and geometry, this sensor consists of a steel cylinder (height: 2.25 mm, diameter: 5.8 mm) filled with a gel membrane protecting the sensing element. The latter, together with the cylinder, is glued to a ceramic printed circuit board (PCB) (6.4 × 6.2 × 0.63 mm), which contains the pads for the electrical connection.

To power the transducer and acquire data, we prepared two multi-pole cables made of enameled copper wire (0.10 mm dia.) for electric motor windings. The first step consisted of cutting the spool of copper wire into strands of 50 cm. For each sensor, four strands were needed: power, ground, positive output, and negative output. To distinguish the wires, their extremities were colored in red, black, white, and green, respectively. To distinguish between the six sensors, we first divided them into two triads identified by a colored shrinking tube heated on their wires: white and yellow. Similarly, we identified each of the sensor in the triad with the following colors: red, green, and blue representing the X, Y, and Z axes, respectively.

The strands of each triads were wrapped around a fishing line made of Dyneema (diameter: 0.30 mm, tensile strength: 230 N) to increase the strength of the cable. Sensors soldered to this cable are shown in Figure 2.

### 2.4. Construction of the MAPS Probe

To create and test the MAPS probe, we decided to embed the pressure sensors inside a cubic specimen of 50 × 50 × 50 mm sides. To guarantee a good alignment of the sensors, we decided to divide the casting procedure into three steps, and we created two molds using 3D-printed casts made of polylactic acid (PLA) to the purpose. The strategy adopted consisted of a first casting of the upper part of the cube inside a rectangular mold, which contained the positive imprint of the sensors. After the first half of the cube was demolded, sensors were placed inside the cavity obtained in the rubber from the first cast, following the scheme shown in Figure 1. Before placing the sensors, those were coated with some uncured material to allow the adhesion of the sensor with the cured rubber of the specimen. After the sensors were positioned and the applied material was cured, the second mold, consisting of a rectangular extrusion, was positioned. We prepared this mold such that the cables could exit on one of the vertices of the specimen. After positioning, the upper mold was flooded with uncured material and left to cure. Molds are shown in Figure 3, and casting steps are shown in Figure 4.

### 2.5. Pressure Sensor Uniaxial Test

To verify the feasibility of using piezoresistive pressure sensors in our test probe, we decided to conduct a brief preliminary study on a single pressure sensor to evaluate its uniaxial response to external loads when embedded in a silicone rubber. To do so, we prepared a cylindrical specimen (diameter: 25 mm, height: 50 mm) with a pressure sensor embedded in its center. This cube was put inside a tensile testing machine and tested in cyclic compression and tension tests by applying a series of ten sine waves at 1 Hz frequency and 20% strain range. The aim of this pilot analysis was twofold: to verify if the pressure sensor was suitable for tensile measurements and to check its linearity. The preparation and calibration procedure of the uniaxial pressure sensor is shown in Figure 5.

### 2.6. Calibration of the MAPS Probe

To calibrate the MAPS probe, we used a MiniBionix II (MTS) testing machine. This machine is actuated by a servo hydraulic cylinder (stroke: 100 mm) and is equipped with a linear variable displacement transducer (LVDT) sensor (range: 100 mm) and a load cell (full scale: 1.5 kN) to control and measure displacement and force, respectively. The software allows the user to move the cylinder with custom displacement or force profiles. In the calibration trials, we decided to apply a series of ten sine waves (amplitude: 10 mm, frequency: 1 Hz), which corresponded to a 20% deformation of the cube. To conduct testing both in shear and compression, we designed custom grips to mount the MAPS probe to the machine. In the compression trials, two large aluminum plates were fixed to the machine clamps. The cube was then located on the lower plate, and the upper plate was lowered until it was completely touching the upper surface of the cube. In the shear tests, we added to the machine two vertical and parallel plates, the width of which could be adjusted to match the thickness of the specimen. In both cases, the stickiness of the material was sufficient to guarantee the adhesion of the specimen to the surfaces of the grips, which was crucial in the shear tests. The MAPS probe mounted on the grips and ready for the tests is shown in Figure 6.

We conducted tests in compression along the three axes X, Y, and Z. Then, we conducted shear tests in the three directions XY, YZ, and XZ. To acquire the data, we used a SoMat eDaqLITE (HBM) datalogger equipped with two ELBRG modules, which implement strain gauges signal conditioners. The acquisitor was also connected to the output channels of the testing machine to collect the displacement and force signals synchronously with the pressure sensors output. Data were collected with a sampling rate of 100 Hz.

To analyze the data, we used the software MATLAB R2020b (MathWorks). First, data were filtered using a lowpass filter with cut-off frequency of 2 Hz. Then, we calculated the nominal pressure p_nom_ applied to the specimen by dividing the force by the nominal area of the specimen, equal to 2.5 cm^2^. This pressure was equal to the opposite of the σ during the compression test and equal to the τ during the shear test. Using the p_nom_ and the output of the pressure sensors, we calculated the sensitivity matrix **S** of the MAPS probe. To fill this matrix, we computed each of the s_ij_ terms as the linear regression between data from the sensor outputs and the nominal load, as shown in Figure 7.

We completed the six columns using the compression on the X, Y, Z axes and the shear tests in XY, YZ, XZ.
(21)$$\left[\begin{array}{c} p_{1}\\ p_{2}\\ \begin{array}{c} p_{3}\\ p_{4}\\ \begin{array}{c} p_{5}\\ p_{6} \end{array} \end{array} \end{array}\right]=\left[\begin{array}{cccccc} s_{p_{1},\sigma_{x}} & s_{p_{1},\sigma_{y}} & s_{p_{1},\sigma_{z}} & s_{p_{1},\tau_{xy}} & s_{p_{1},\tau_{yz}} & s_{p_{1},\tau_{xz}}\\ s_{p_{2},\sigma_{x}} & s_{p_{2},\sigma_{y}} & s_{p_{2},\sigma_{z}} & s_{p_{2},\tau_{xy}} & s_{p_{2},\tau_{yz}} & s_{p_{2},\tau_{xz}}\\ s_{p_{3},\sigma_{x}} & s_{p_{3},\sigma_{y}} & s_{p_{3},\sigma_{z}} & s_{p_{3},\tau_{xy}} & s_{p_{3},\tau_{yz}} & s_{p_{3},\tau_{xz}}\\ s_{p_{4},\sigma_{x}} & s_{p_{4},\sigma_{y}} & s_{p_{4},\sigma_{z}} & s_{p_{4},\tau_{xy}} & s_{p_{4},\tau_{yz}} & s_{p_{4},\tau_{xz}}\\ s_{p_{5},\sigma_{x}} & s_{p_{5},\sigma_{y}} & s_{p_{5},\sigma_{z}} & s_{p_{5},\tau_{xy}} & s_{p_{5},\tau_{yz}} & s_{p_{5},\tau_{xz}}\\ s_{p_{6},\sigma_{x}} & s_{p_{6},\sigma_{y}} & s_{p_{6},\sigma_{z}} & s_{p_{6},\tau_{xy}} & s_{p_{6},\tau_{yz}} & s_{p_{6},\tau_{xz}} \end{array}\right]{\left[\begin{array}{c} \sigma_{x}\\ \sigma_{y}\\ \begin{array}{c} \sigma_{z}\\ \tau_{xy}\\ \begin{array}{c} \tau_{yz}\\ \tau_{xz} \end{array} \end{array} \end{array}\right]}^{nom}$$

After completion of the **S** matrix, its inverse **C** was computed.
(22)$$C=S^{-1}$$
(23)$${\left[\begin{array}{c} \sigma_{x}\\ \sigma_{y}\\ \begin{array}{c} \sigma_{z}\\ \tau_{xy}\\ \begin{array}{c} \tau_{yz}\\ \tau_{xz} \end{array} \end{array} \end{array}\right]}^{est}=C\left[\begin{array}{c} p_{1}\\ p_{2}\\ \begin{array}{c} p_{3}\\ p_{4}\\ \begin{array}{c} p_{5}\\ p_{6} \end{array} \end{array} \end{array}\right]$$

### 2.7. Static Validation of the MAPS Probe

After the calibration of the MAPS probe, we performed a bi-axial static load test to verify the quality of the calibration matrix and to quantify the cross-sensitivity of our stress probe. Similarly to what other researchers did to test their shear sensors [18], we built a test bench using 3D-printed components made of PLA to apply a compressive load plus a shear load.

The compressive load was applied thanks to a cylinder, which had housing for placing calibrated weights and was constrained to slide along a sleeve. To apply the shear load, the sleeve was mounted on two linear guides and pulled by a wire trough a pulley system. Finally, to test the two directions of shear on the same plane, a rotating base was located over the base of the frame, allowing us to test multiple shear directions without repositioning the specimen. A sketch indicating the components of the test bench and a photo of it during a test are shown in Figure 8. The described test bench allowed us to compress the cube with a weight of 500 g, while also applying a shear load with a weight of 100 g. These two loads corresponded to a σ^nom^ equal to −1.96 kPa and a τ^nom^ equal to 0.39 kPa. When the two loads were applied together, the nominal σ_id,VM_ was equal to 2.07 kPa.

## 3. Results

### 3.1. Uniaxial Pressure Sensor Test

Compression and tension tests conducted on the cylindrical specimen containing the pressure sensor showed linearity and sensibility to both compressive and tensile stresses. Indeed, we found high R^2^ values obtained in both tests, which confirmed the validity of our linear fitting. Moreover, we found that pressure sensor output was close to the nominal pressure applied, with a sensitivity coefficient of 0.94 in compression and of 1.18 in tension. Results are summarized in Figure 9.

### 3.2. Calibration of the MAPS Probe

Examples of raw outputs of the sensor during a compression and a shear calibration trial are shown in Figure 10. The resulting sensitivity coefficients of the matrix S are reported in Table 1. The R^2^ values of the fitting are reported in Table 2.

The sensitivity coefficients showed that during the compression trials, the aligned sensor was the most sensible to the applied load; however, the sensor output was 1.5–1.77 times more than the nominal pressure applied to the cube. The transversal sensors instead were sensing a slight tension, which may be due to the Poisson effect. Indeed, the bulkiness of the specimen may have constrained the lateral expansion of the material, causing the sensed stress. During the shear tests, the auxiliary sensor acting on the plane of shear was the most sensitive and sensed about 2.7 times the nominal load applied. In terms of goodness of the fit, most of sensitivity coefficients showed high values of R^2^ (>0.9). The sensitivity matrix was invertible, as its determinant was equal to 146. The calibration matrix **C** resulting from the inversion is reported in Table 3. This matrix was dense, with the highest terms on the diagonal and non-zero entries in the other positions.

### 3.3. Static Validation of the MAPS Probe

The quality of the calibration matrix was verified using the test bench described in the methods section, which we used to conduct a pilot test. Thanks to the calibration matrix, we were able to compute the stress components by pre-multiplying the vector of pressure sensor output by **C.** In Figure 11, the estimated stress state obtained is compared to the nominal external loads applied with the weights. When the specimen was compressed in X while also applying a shear stress in Y or in Z direction, the MAPS probe was able to recognize the applied stresses with a maximum error of 4.7% in compression, 7.5% in shear XY, and 13.8% in shear XZ with respect to the nominal loads applied. The σ_id,VM_ was estimated by the probe with an error of 5.8% in the two combined load conditions. To quantify the cross-sensitivity, we analyzed the estimated stress components, which were not directly loaded during the tests. The highest cross-sensitivity was measured during the shear XZ trial, where a τ_xy_ of 0.12 kPa (30% of the nominal load τ^nom^_xz_) was measured by the MAPS probe.

We also computed the stress components that we obtained by the analytical solution described in the methods, which are shown in Figure 12. Qualitatively, the applied stresses were identified correctly; however, those were amplified when compared to the nominal load applied to the specimen resulting in higher errors with respect to the stress state estimated by using the calibration matrix, and a higher cross-sensitivity was found. As a consequence, the σ_id,VM_, which was calculated using the analytical approach, was 85% higher with respect to the σ_id,VM_ nominal.

## 4. Discussion

Results of the compression and tension tests conducted on the uniaxial pressure sensor specimen have shown that the chosen transducer is able to measure compression and tension with a linear response. These preliminary observations that were confirmed by the calibration trials conducted on the MAPS probe evidence a good sensitivity of the pressure transducers both in compression (during the compression tests) and in tension (during the shear tests), indicating a good adhesion of the sensors to the rubber of the specimen and the capability of the sensors to detect negative pressures when embedded in a solid material.

We analyzed the raw sensor data to extract their relationship with the nominal load applied to the specimen. When the specimen was compressed with 20% strain, we measured a nominal pressure within the range of 0 to 6 kPa in compression and 0 to 0.5 kPa in shear.

Qualitatively, pressure sensor output agreed with the expected output given by the analysis of stress and the position of the sensors. Indeed, diagonal terms of **S** were the highest, meaning that in each load case sensor aligned to the principal direction of stress was sensing more than the others. However, when looking in detail at the elements of the sensitivity matrix **S** of Table 1, we found an unexpected over estimation of the nominal load, ranging from 1.5 to 1.77 in compression trials and up to 2.97 in shear trials. Nevertheless, the linear fitting proved good in terms of R^2^, which was more than 0.9 for most of the sensitivity coefficients. Low values of R^2^ were corresponding mainly to shear tests where the lower load applied decreased the signal to noise ratio of our measurement apparatus and caused a slightly higher dispersion of data. After these trials evidenced the little stress needed to achieve high deformation of this material, precision could be improved by adopting sensors with a lower full-scale output than 700 kPa.

Sensitivity values resulted from the computation of **S** matrix supported the need of introducing the calibration matrix **C** (Table 3) to correct the raw outputs of the pressure sensors and obtain an accurate estimation of the applied loads. The calibration matrix was dense, indicating that all sensors contributed to the definition of stress components, though its diagonal terms were again the highest in modulus meaning that for each direction the sensor positioned in the direction of the principal stress was contributing more than the others, as expected by the analytical model.

Validation trials were conducted using a static test bench built for the purpose. This approach, though it did not evaluate the dynamic response of the sensors, allowed us to test up to two load directions simultaneously and was a validation tool, which was also successfully adopted in past research [18,19]. This validation trial was crucial to the research because the final aim of the work was to estimate the stress state inside the material, which is composed of six independent components. Though we were not able to set up a triaxial test bench, our bi-axial validation test allowed us to verify that MAPS probe was competitive with other planar stress sensors for soft materials [16,17,18,19]. Differently to these sensors, the position of the sensors and the geometry of our probe replicates in three directions, making the findings of the biaxial test also applicable to the other planes: we could, therefore, expect that the MAPS should also work well in the measurement of triaxial stress states.

The bi-axial test bench allowed us to apply 1.96 kPa of compressive stress, together with a shear stress of 0.39 kPa, giving a combined σ_id,VM_ of 2.07 kPa. The nominal load was compared with the experimental stress state computed both with the calibration matrix and the analytical solution of the equations of stress. The calibration matrix approach was effective in detecting separately the compressive and shear stresses, resulting in a maximum error of 5.8% in estimating the σ_id,VM_. Most of the error came from the shear stress in XY direction where we found a high cross-sensitivity of 0.12 kPa (corresponding to 30% of the nominal load in the XZ direction). Conversely, the analytical solution gave higher error (the estimated σ_id,VM_ was 85% higher than the nominal) and cross sensitivity. High errors of the analytical solution may derive from sensor misalignment as well from intrinsic cross-sensitivity of the pressure transducer itself due to the disturbance of the gel behavior.

Hence, given the low errors obtained in the calibration and validation trials, we could hypothesize that the MAPS probe could be successfully used in a full-scale stress measurement of soft materials such as helmet testing if embedded inside a surrogate brain made of silicone rubber. Physical head models, which have a soft brain surrogate [3,4,5,6], could take advantage of this sensor to improve its measure capability. Indeed, the MAPS could be used to effectively determine the VMS, which could be compared with a threshold to assess the risk of TBI [9,10]. Furthermore, the stress state can be used to determine the strain state by linking these two entities with a material model of the surrogate, such as the one described in Appendix A. Knowing the strain state is crucial to calculating axonal strain and, thus, to assessing the impact conditions, which could generate a DAI [10]. Moreover, the measure of the stress state obtained thanks to the placement of MAPS probes at known and relevant locations possibly suggested by FE models could be compared with the output of these models [7,8], serving as a cross-validation tool. Moreover, the comparison with FE models could help to the definition of an experimental VMS threshold, which could be compared with the MAPS output. Oeur et al. reported the peak VMS of their simulations of accidents with different TBI outcome, which varied from 8.2 kPa (subdural hematoma) to 15.4 kPa (persistent postconcussive symptoms) [21]. Thanks to simulations from other authors, VMS thresholds relative to a 50% risk of TBI and ranging from 27 to 61.6 kPa were proposed [9,10]. However, the outputs of these simulations would be hard to compare with the MAPS probe output, since they are based on peak values of the FE model and are likely very localized and located at the interfaces of the brain with the other anatomical structures, where stresses are higher. Conversely, the MAPS probe will mainly be suitable to measure stress states inside the brain, and its capability of application at material interfaces has yet has to be investigated.

We promote our MAPS probe, since it is cheap and relatively easy to be produced when compared to other stress sensors. Indeed, the preparation of our transducers requires only the good soldering of wires on the PCB of the sensor, making it simpler than other approaches, which need to connect wires to a rubber transducer [11,12,13,14]. Moreover, it is also more informative than planar stress sensors, as it adds the capability of measuring the triaxial stress state [16,17,18,19]. Furthermore, the MAPS probe could potentially be embedded in a large variety of soft materials with no modifications and would work well in a softer or stiffer rubber, increasing the field of possible applications for this type of probe. Finally, the working principle is independent to the pressure transducer adopted, and strain sensors such the ones based on PDMS could be investigated as possible substitutes for the pressure transducers to use the hexadic arrangement of the MAPS probe to create a triaxial strain sensor [11,12,13,14].

Future work on the MAPS probe will aim to address the main limitations evidenced during this work. Results of the calibration trials will be furtherly investigated to improve the validity of the analytical solution. Indeed, results have evidenced the need to use the MAPS probe only after an experimental calibration and to associate a calibration matrix to the sensor, making the preparation of MAPS more time consuming. More tests will also investigate the repeatability of the probe, the repeatability of its calibration matrix, and the dynamic response. Alternative pressure or strain transducers with reduced size will be explored to reduce the distortion caused by the sensors and to improve its spatial resolution. Finally, a dynamic evaluation test will be studied to assess the dynamic response of the sensor under multi-axial loads. Full-scale tests of our head surrogate equipped with the MAPS will be performed to verify the functioning of the probe during impacts and to guide its development towards the optimal size and sensing range.

## Figures and Tables

**Figure 1 sensors-21-03487-f001:**
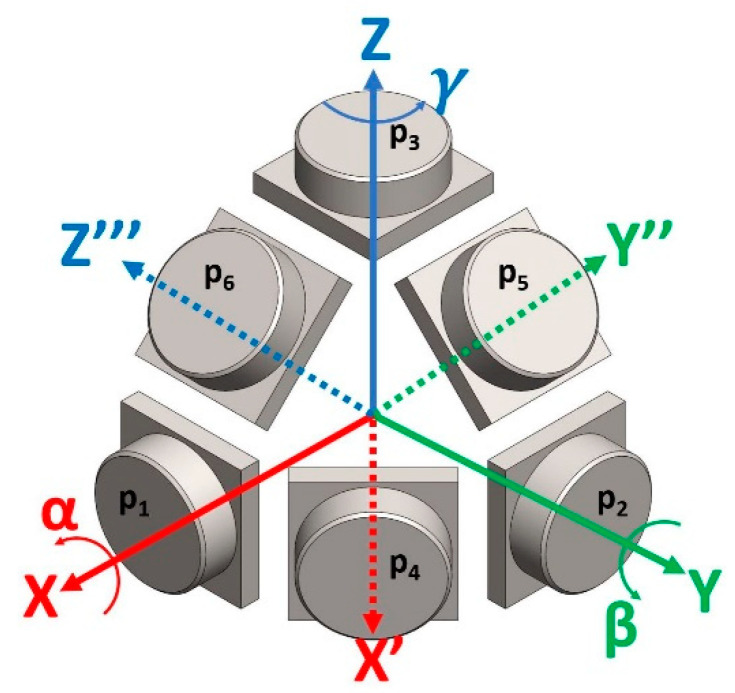
MAPS probe hexad pressure sensor disposition: the six sensors are disposed in a cartesian triad XYZ plus the bisectors of the first triad axes X′, Y′′, and Z′′′.

**Figure 2 sensors-21-03487-f002:**
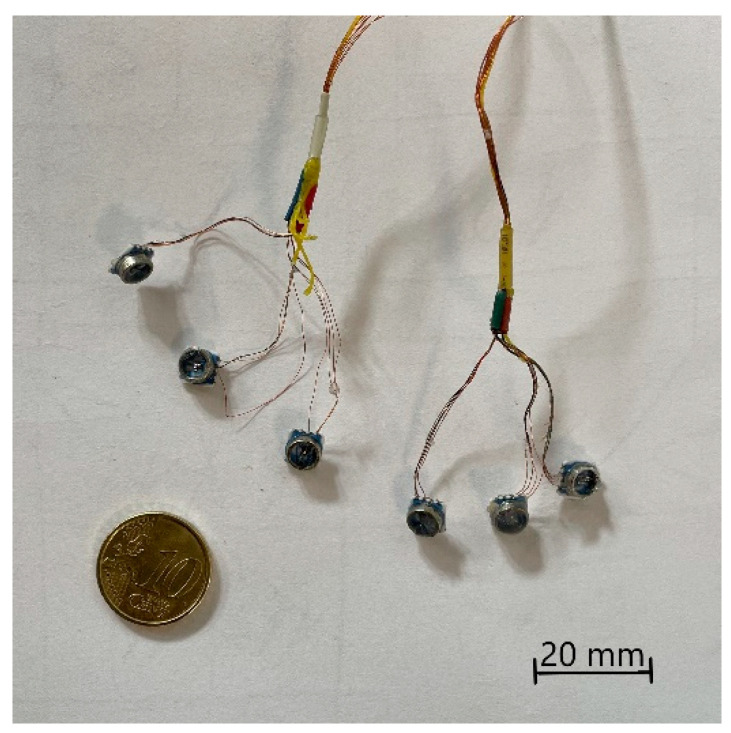
MS5407 pressure sensors soldered to the cables, shrinking tube uniquely identifies each sensor.

**Figure 3 sensors-21-03487-f003:**
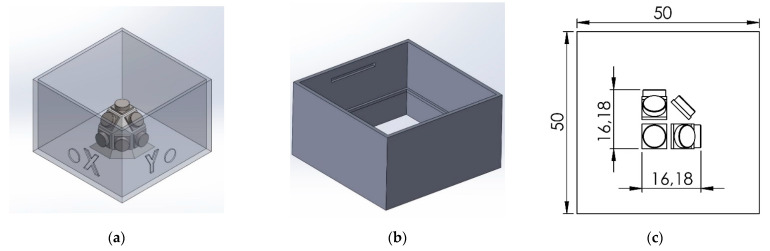
Casting molds: (**a**) Lower mold with the positive print used to create the cavity for the sensor positioning, marks are printed to the surfaces to indicate the sensor position and identify the cube axes. (**b**) Upper mold used to complete the cube, marks on the surfaces allow for alignment of the mold over the first half of rubber specimen. (**c**) Position and size of the stress probe inside the testing specimen, the volume occupied by the actual sensing probe is confined to a cube of 17 mm side.

**Figure 4 sensors-21-03487-f004:**
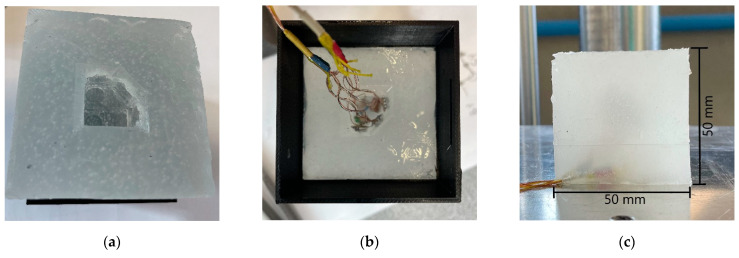
Casting procedure: (**a**) Result of the first cast, the upper part of the specimen is realized with a cavity to insert the sensors. (**b**) Sensor positioned inside the cavity; uncured silicone rubber is added to make the sensor adhering to the cured material. The upper cast is positioned for the final step. (**c**) Result after the casting, sensors are embedded, and their cables exit from a vertex of the specimen.

**Figure 5 sensors-21-03487-f005:**
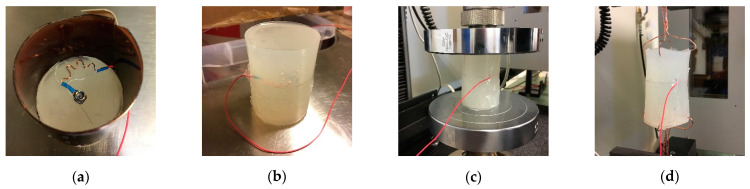
Preparation and testing of the uniaxial pressure sensor: (**a**) Pressure sensor positioned over the first half of the rubber specimen. (**b**) Cylindrical specimen ready for the test. (**c**) Compression test. (**d**) Tension test.

**Figure 6 sensors-21-03487-f006:**
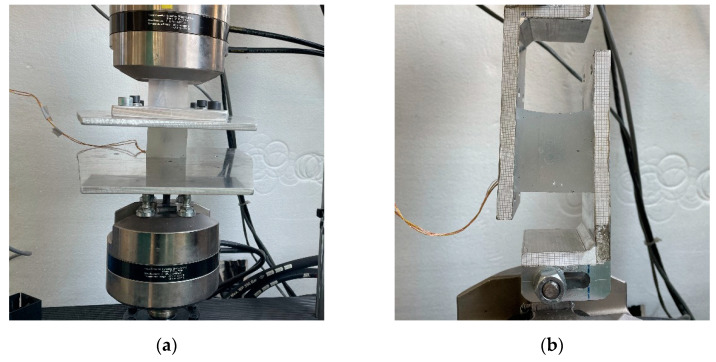
Test conditions: (**a**) Cube positioned for compression tests. (**b**) Cube positioned for shear tests.

**Figure 7 sensors-21-03487-f007:**
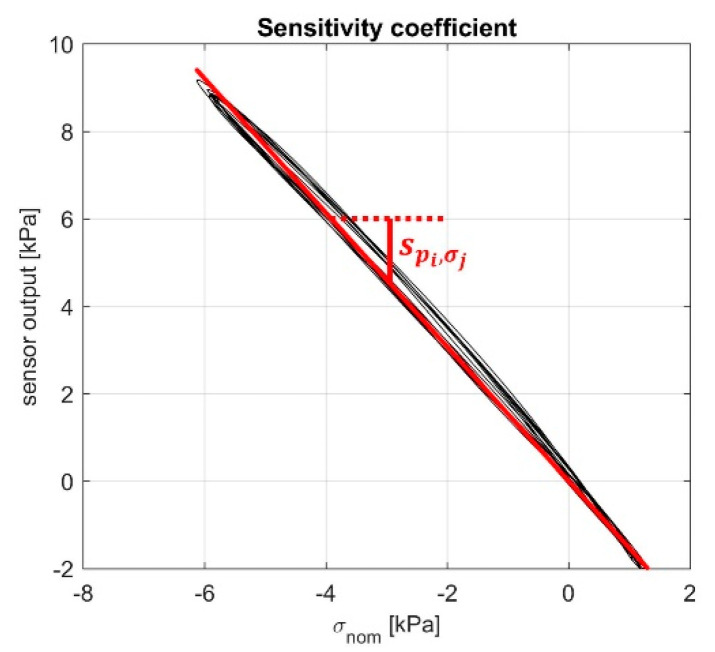
Example of the computation of sensitivity coefficients.

**Figure 8 sensors-21-03487-f008:**
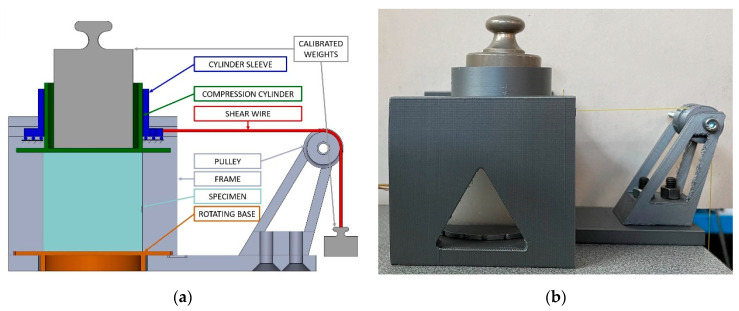
Test bench for the multi-axial evaluation of the sensor: (**a**) Sketch with indication of the components. (**b**) Actual test bench during a compression and shear test.

**Figure 9 sensors-21-03487-f009:**
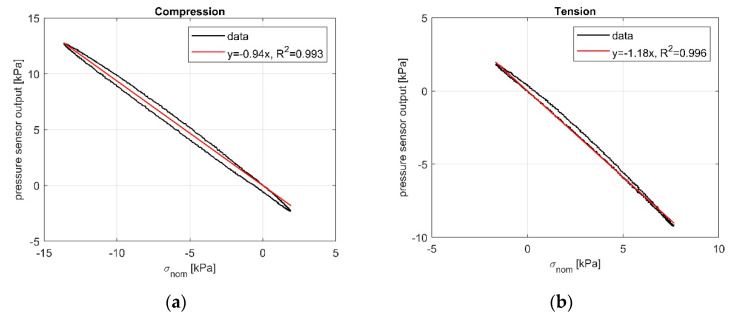
Uniaxial cyclic pressure sensor tests: (**a**) Compression. (**b**) Tension.

**Figure 10 sensors-21-03487-f010:**
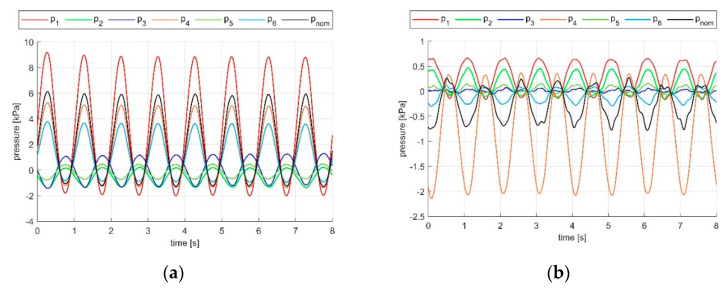
Raw outputs of sensors compared to the nominal load applied (in black): (**a**) Compression test along X. (**b**) Shear in direction XY.

**Figure 11 sensors-21-03487-f011:**
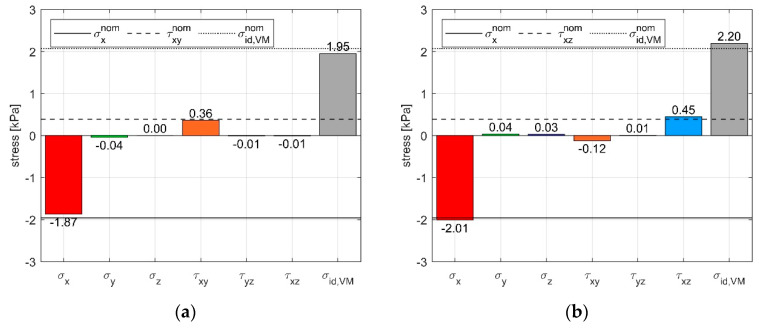
Response of the stress probe under combined loads: (**a**) Compression on X plus shear in XY direction. (**b**) Compression on X plus shear in XZ direction.

**Figure 12 sensors-21-03487-f012:**
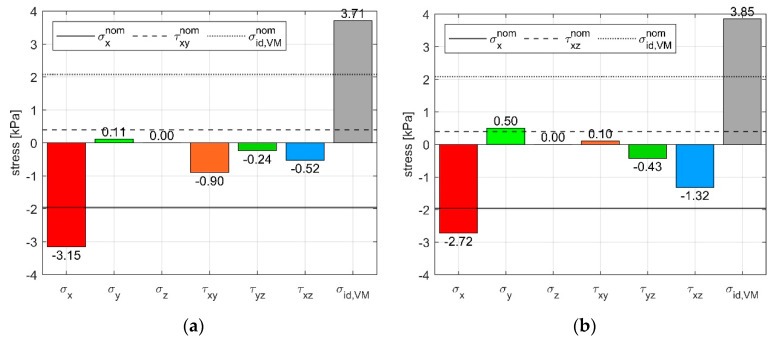
Shear stress computed from the analytical solution during combined loads: (**a**) Compression on X plus shear in XY direction. (**b**) Compression on X plus shear in XZ direction.

**Table 1 sensors-21-03487-t001:** Sensitivity coefficients: ratio between each sensor output (positive in compression) and nominal load (positive in traction) for the six tested load conditions.

Sensor ID	Orientation	X	Y	Z	XY	YZ	XZ
*p* _1_	X	−1.518	0.032	0.142	0.870	0.044	−0.529
*p* _2_	Y	0.212	−1.629	0.206	0.599	−0.236	0.117
*p* _3_	Z	0.343	0.284	−1.772	0.036	1.109	0.971
*p* _4_	X′	−0.891	−0.686	0.019	−2.679	−0.557	−0.839
*p* _5_	Y′′	0.162	−0.882	−0.821	0.164	−2.741	0.546
*p* _6_	Z′′′	−0.634	0.230	−1.398	−0.388	−0.233	−2.798

**Table 2 sensors-21-03487-t002:** R^2^ index of the sensitivity matrix coefficients of Table 1.

Sensor ID	Orientation	X	Y	Z	XY	YZ	XZ
*p* _1_	X	0.997	0.839	0.994	0.952	0.246	0.943
*p* _2_	Y	0.995	0.996	0.990	0.942	0.731	0.652
*p* _3_	Z	0.992	0.995	0.994	0.207	0.961	0.959
*p* _4_	X′	0.999	0.998	0.605	0.973	0.959	0.972
*p* _5_	Y′′	0.987	0.998	0.998	0.795	0.979	0.943
*p* _6_	Z′′′	0.999	0.983	0.999	0.978	0.949	0.977

**Table 3 sensors-21-03487-t003:** Calibration matrix obtained by inverting the matrix S reported in Table 1.

	p_1_	p_2_	p_3_	p_4_	p_5_	p_6_
σ_x_	−0.614	0.090	−0.125	−0.202	−0.038	0.129
σ_y_	−0.017	−0.596	−0.089	−0.137	0.043	−0.003
σ_z_	−0.036	−0.011	−0.383	−0.006	−0.141	−0.152
τ_xy_	0.164	0.113	−0.038	−0.301	0.034	0.058
τ_yz_	0.016	0.188	0.174	0.031	−0.316	−0.006
τ_xz_	0.132	−0.095	0.203	0.077	0.104	−0.318

## Data Availability

Not applicable.

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
