# Peer review of "A Novel Multi-Axial Pressure Sensor Probe for Measuring Triaxial Stress States Inside Soft Materials"

_sensors, 2021, doi:10.3390/s21103487_

Round 1

Reviewer 1 Report

As the author said that the measurement of triaxial stress states inside bulk materials such as brain tissue surrogates is a challenging task. But the presented multi-axial pressure sensor probe is too big volume to be suitable for application in brain tissue surrogates. Has the author considered how to reduce its volume?  Which would make the research more meaningful.

Author Response

We sincerely thank the reviewer for having appreciated our work. We will proofread the manuscript more carefully to improve the language and style as the reviewer suggested.

Regarding the specific comment we managed to create a probe which is 17x17x17 mm size after its extraction from the testing specimen (50x50x50 mm) used to conduct the calibration and validation trials exposed in the paper. Thus, the size of the probe allows its placement in at least 3 or 4 locations inside our brain surrogate for impact testing. However, we still want to increase the spatial resolution of our MAPS and thus we are looking at alternative sensors to reduce furtherly its size such as digital pressure sensors which could easily half its size (e.g., MS5840-02BA by TE connectivity), but we are still searching from a small sensor with appropriate range and output data rate. Nevertheless, the MAPS probe we created was a proof that the geometrical disposition we thought was effective to the purpose. We are also confident that the probe we presented could improve the measurement capability of our instrumented brain surrogate by giving an estimation of stress even if it is averaged to the size of the probe. We added a sentence to the discussion to inform the reader about this limitation and future work (line 451-452): “Alternative pressure or strain transducers with reduced size will be explored to reduce the distortion caused by the sensors and to improve its spatial resolution.”

Reviewer 2 Report

This paper introduces a Multi-Axial Pressure Sensor (MAPS) based on the hexadic disposition of six sensors to measure triaxial stresses inside soft materials. Its target application is brain kinematics during impacts. The implementation stage is comprehensively described together with stress compression/shear behavior characterization.

Overall, this work presents a novel and interesting geometric approach for triaxial stress sensors. The paper is easy to read and to follow. Nevertheless, I recommend a major revision expecting to see the following remarks addressed in the revised version:

A. Content

- It is mentioned that the proposed MAPS could help in the development of head protective gear as it measures the stresses that the head might suffer during a trauma. However, the sensing ranges (3 to 9 kPa in fig 10) are no longer related to the intended application. How are these values suitable for the application?

- The Von Misses stress definition is superficially overviewed. Why is it relevant to the characterization of the development?

- Figure 11: y label missing

- The electrical characterization of the sensor is not given. How are the pressure values translated into voltage? It would be interesting to know if the voltage follows a linear behavior or if there is a hysteresis phenomenon.

B. Format

- Line 13: Purpose of the work --> The purpose of the work

- Line 36: This sentence is confusing. Rephrase to: However, despite the number of physical surrogates already developed [3–6],

- 58: ROM not defined

- 75: it’s --> it is

- 110: Incorrect format for section 2. Subsection 2.1 is in the same line

- 152: system of three with three variables --> system with three variables

- 414: DAI not defined

- The indentation for the figures and their labels must be the same across the entire paper

Author Response

We thank the reviewer for having appreciate the effort we spent in developing and testing our MAPS probe. We are also glad that he/she found the paper clear. We did our best to address each of the reviewer’s comments to improve the quality of the work.

  1. In the calibration stage of the MAPS probe we applied a 20% strain with a 1 Hz frequency, we decided to not exceed these values to reduce non-linear factors such as geometrical distortion of the initial sensor disposition and material behavior. Indeed, these first trials were aimed to assess the feasibility of the MAPS probe. However, we are aware that increasing the strain and the rate could lead to higher ranges of measured pressure, bringing them closer to the intended application. We found in literature a Von Mises Stress threshold for assessing TBI (50% risk) ranging from 27 (Marjoux et al., 2008) to 61.6 kPa (Sahoo et al., 2016). Moreover, other simulations (Oeur et al., 2015) showed a peak VMS closer to our sensing range and ranging from 8 kPa (subdural hematoma group) to 15.4 kPa (persistent postconcussive symptoms group), which are not extremely far from the values we have reached in our calibration trials. Nevertheless, the values presented above were peak values from FE models while our probe will likely provide a less localized measure. Moreover, these peak values are likely close to the brain interfaces with other anatomical structures where stress could be possibly higher: on the other hand, our probe will be mainly suitable to measure stress states inside the brain and its capability of application at material interfaces has yet to be investigated. Future work will aim to include the MAPS in our physical head brain surrogate and to perform some impact tests. The data obtained will be used to tune the sensing range of future versions of the MAPS and will work in pair with material choice and testing to improve step wisely the bio-fidelity of our surrogate. We added this consideration in the text. Lines 425-436: “Moreover, the comparison with FE models could help to the definition of an experimental VMS threshold which could be compared with the MAPS output. Oeur et al. reported the peak VMS of their simulations of accidents with different TBI outcome, which varied from 8.2 kPa (subdural hematoma) to 15.4 kPa (persistent postconcussive symptoms) [21]. Thanks to simulations from other authors, VMS thresholds relative to a 50% risk of TBI and ranging from 27 to 61.6 kPa were proposed [9,10]. However, the outputs of these simulations would be hard to compare with the MAPS probe output since they are based on peak values of FE model and are likely very localized and located at the interfaces of the brain with the other anatomical structures, where stresses are higher. Conversely, the MAPS probe will be mainly suitable to measure stress states inside the brain and its capability of application at material interfaces has yet has to be investigated.” Lines 458-460: “Full scale tests of our head surrogate equipped with the MAPS will be performed to verify the functioning of the probe during impacts and to guide its development towards the optimal size and sensing range.”

  2. We included the Von Mises stress in our study because it represents a useful parameter to condense the complex information of a triaxial stress state in a single value. In the development of our probe was useful to quantify the global accuracy of our probe. Moreover, it is a meaningful parameter to quantify stress and to compare it with a threshold which could be linked with TBI in the specific case. We added this in two sentences in the text to clarify. Lines 162-164: “With this set of equations, the stress in the point of application of the probe can be fully determined in its components, and eventually condensed in the Von Mises Stress (VMS) parameter to compare the stress intensity with a threshold.” Lines 417-419: “Indeed, the MAPS could be used to determine effectively the VMS, which could be compared with a threshold to assess the risk of TBI [9,10].”

  3. We thank the reviewer for the observation, the label “stress [kPa]” has been added to the charts of Figure 11.

  4. Typical and worst linearity of the sensors stated by the manufacturer are ±0.15% and ±0.40%, correspondingly. This parameter was tested and appeared below 1%. Larger problem is the large sensitivity scatter stated by the manufacturer. Nevertheless, the tests which we did not include for brevity have shown that the sensitivity differences for the sensors from the same batch lay within few %, which was regarded as acceptable in the situation with prior calibration of the sensors before encapsulating them into the MAPS. We added some more details in the text, now it reads (lines 178-180): “This sensor offers a full-scale pressure of 700 kPa and transmits data through a differential analog output (full scale output: 392 mV, 5 V power supply), according to the producer’s specifications linearity is ±0.15% (typical) of the full-scale pressure.”

  5. We corrected the typo

  6. We thank the reviewer for the suggestion, we rephrased the sentence accordingly. Now it reads (line 36): “However, despite the number of physical surrogates already developed [3–6], …”.
  7. We thank the reviewer for the observation, we modified the word to “motion”.

  8. We corrected the typo

  9. We corrected the typo
  10. We thank the reviewer for the comment, now it reads (line 153): “The three equations above form a system with three variables: …”

  11. We thank the reviewer for the comment, we missed to define the acronym in line 43. Now it reads (lines 41-44): “A measure of brain stress and strain could improve the experimental methods to test the effectiveness of head protective gear since these quantities are correlated with Traumatic Brain Injury (TBI) such as Diffused Axonal Injury (DAI) a major cause of death after traumatic events [9,10].”

  12. To prepare our manuscript we followed the template provided by the journal, in this template the figure with multiple panels were occupying the whole page while the single panel figures were indented in line with the main text. Nevertheless, we did find and correct some inaccuracies and we corrected them: indentation of figure 5 (line 236-237), caption centering of figure 9 (line 305).

Reviewer 3 Report

In this paper, the authors developed and characterized a device for measuring triaxial stress states inside bulky soft materials based in six pressure sensors. The work is clear and well-structured, and I can recommend this work for publication after the following revision:

  • In line 110, Place the titles on different lines.
  • Please indicate the error bars in the values of tables 1 and 3.
  • The authors should clearly identify in the discussion section what are the advantages of this device compared to others already presented in the literature.
  • The Author Contribution session is unfilled.

Author Response

We thank the reviewer for its positive feedback on our work. We are also glad that he/she found the paper clear. We did our best to address each of the reviewer’s comments to improve the quality of the work.

  1. We thank the reviewer for the comment, we corrected the typo

  2. The values of table 1 (and consequently of table 3 which contains the inverse) are the coefficients of the regression of data thus they are without error. A measure of error (in terms of how much the sensitivity coefficients reported in table 1 are representative of the sensor output) is given by the R2 values of table 2 instead.

  3. We thank the reviewer for the comment, we added the advantages of our device in the discussion. Now it reads (lines 437-448): “We promote our MAPS probe since it is cheap and relatively easy to be produced when compared to other stress sensors. Indeed, the preparation of our transducers requires only the good soldering of wires on the PCB of the sensor, making it simpler than other approaches which need to connect wires to a rubber transducer [11–14]. Moreover, it is also more informative than planar stress sensors as it adds the capability of measuring the triaxial stress state [16–19]. Furthermore, the MAPS probe could potentially be embedded in a large variety of soft materials with no modifications and would work well in a softer or stiffer rubber, increasing the field of possible applications for this type of probe. Finally, the working principle is independent to the pressure transducer adopted, and strain sensors such the ones based on PDMS could be investigated as possible substitutes for the pressure transducers to use the hexadic arrangement of the MAPS probe to create a triaxial strain sensor [11–14].”

  4. We thank the reviewer for the comment, we changed the text accordingly. Now it reads (lines 463-468): “Author Contributions: Conceptualization, Giuseppe Zullo, Andrey Koptyug and Nicola Petrone; Data curation, Giuseppe Zullo, Anna Leidy Silvestroni and Gianluca Candiotto; Funding acquisi-tion, Nicola Petrone; Investigation, Giuseppe Zullo, Anna Leidy Silvestroni and Gianluca Candiotto; Methodology, Giuseppe Zullo and Nicola Petrone; Resources, Nicola Petrone; Supervision, Andrey Koptyug and Nicola Petrone; Validation, Giuseppe Zullo; Writing – original draft, Giuseppe Zullo and Anna Leidy Silvestroni; Writing – review & editing, Andrey Koptyug and Nicola Petrone.”

Round 2

Reviewer 2 Report

The paper has undoubtedly improved from its previous version. I appreciate that my remarks were taken into account. In particular, the pertinence of the MAPS’ output values for the intended application gives an idea of its feasibility. 

I have no further major comments or remarks. Therefore, I recommend now the acceptance of this paper.

Reviewer 3 Report

This manuscript has been improved considering the  suggestions and, therefore, I can recommend this work for publication.